# Total Anomalous Pulmonary Venous Return in the Time of SARS-CoV-2—Case Report

**DOI:** 10.3390/children10020387

**Published:** 2023-02-16

**Authors:** Alina-Costina Luca, Alexandrina-Ștefania Curpăn, Raluca-Stefania Manea, Lacramioara Ionela Butnariu, Elena Țarcă, Iuliana Magdalena Starcea, Solange Tamara Roșu, Dana Elena Mîndru, Elena Macsim, Heidrun Adumitrăchioaiei, Ioana Alexandra Pădureț

**Affiliations:** 1Department of Pediatrics, Faculty of Medicine, Grigore T. Popa’ University of Medicine and Pharmacy, 700115 Iasi, Romania; 2Pediatrics Department, “St. Mary” Children’s Hospital, Vasile Lupu Street, No 62-64, 700309 Iasi, Romania; 3Department of Biology, Faculty of Biology, Alexandru Ioan Cuza University, 700505 Iasi, Romania; 4Department of Medical Genetics, Faculty of Medicine, “Grigore T. Popa” University of Medicine and Pharmacy, University Street, No 16, 700115 Iasi, Romania; 5Department of Surgery II—Pediatric Surgery, ”Grigore T. Popa” University of Medicine and Pharmacy, 700115 Iasi, Romania; 6Nephrology Clinic, “St. Mary” Children’s Hospital, Vasile Lupu Street, No 62-64, 700309 Iasi, Romania; 7Emergency Room, “St. Mary” Children’s Hospital, Vasile Lupu Street, No 62-64, 700309 Iasi, Romania; 8Radiology Department, “St. Mary” Children’s Hospital, Vasile Lupu Street, No 62-64, 700309 Iasi, Romania

**Keywords:** TAPVR, SARS-CoV-2

## Abstract

The management of children with complex and life-threatening heart malformations became a clinical conundrum during the SARS-CoV-2 pandemic. The pathophysiological features of the new coronavirus infection have raised major dilemmas regarding the postoperative evolution of an infected patient, and the epidemiological limitations have tightened the criteria for selecting cases. We present the case of a newborn diagnosed with total anomalous pulmonary venous return (TAPVR) who underwent surgical repair of the defect with favorable outcome, despite a prior diagnosis of SARS-CoV-2 infection. We discuss the medical and surgical management of TAPVR, highlighting possible management difficulties brought by the SARS-CoV-2 pandemic.

## 1. Introduction

Under normal circumstances, pulmonary veins drain into the left atrium to ensure the return of oxygenated blood from the lungs to the cardiac chambers and afterwards into the systemic circulation [1], therefore leading to an anomalous pulmonary venous inflow to the right atrium, known as total anomalous pulmonary venous return (TAPVR) [2]. The anomalous pulmonary venous return is a congenital heart disease, accounting for up to 3% of all birth defects; it can be total, with all four pulmonary veins connected to the right atrium either directly or via systemic venous circulation, or partial [3], with at least one pulmonary vein draining in the left atrium [2,4,5]. TAPVR is a cyanotic congenital heart defect that requires an atrial septal defect (ASD) in order for survival to be possible [6].

The Darling classification of TAPVR is based on the site at which pulmonary veins converge [7]:

Type I, supracardiac, is the most common type of all (45–55% of cases): in which the pulmonary veins drain into the innominate vein, the superior vena cava (SVC), or the azygous vein.

Type II, cardiac: the pulmonary veins drain into the right atrium or the coronary sinus. It is generally considered to be benign with postoperative findings concluding that it is associated with a significant degree of pulmonary vein stenosis; 20–30% of patients are diagnosed with this type of TAPVR.

Type III, infracardiac or infradiaphragmatic (25%): a vertical vein serves as a common collector for pulmonary veins; it then drains into the hepatic, portal, or inferior vena cava.

Type IV, the mixed type (5%), involves venous connections at various levels.

The Smith classification takes into consideration the hemodynamic impact of the cardiac anomaly. Non-obstructive TAVPR carries the same consequences as a large atrial septal defect (ASD) [7,8]. The blood passes into the left atrium and into the systemic circulation. However, the atrial septal defect cannot ensure balanced hemodynamics; as a consequence, more blood enters into the right ventricle, leading to right heart hypertrophy, pulmonary congestion, and increased pressure in the right heart cavities and the pulmonary vessels [9,10]. Oxygen levels measured in the aorta and pulmonary artery have the same value due to the mixing of oxygenated and deoxygenated blood in the right atrium [11]. Pulmonary and tricuspid murmurs can be heard, as well as gallop rhythm [8,12].

Obstructed TAPVR leads to pulmonary edema [13]. The obstruction may occur at the interatrial septum or within the anomalous canal: intrinsic narrowing or extrinsic narrowing (in types II and III) [14]. The failure of the pulmonary venous return translates into congestion, increased PVR, and hypertension measured in the right ventricle and pulmonary artery sites. Oxygen levels in the aorta and pulmonary artery have the same value, but remain lower than those registered in patients without obstruction [12].

The selection of the appropriate time for surgical intervention is based on the magnitude of the pulmonary blood flow compared to the systemic blood flow [15]. However, the confirmed diagnosis itself is an indication for surgery.

A ratio between pulmonary (Qp) and systemic (Qs) blood flow greater than 1.5 is a classical indicator for surgical correction in most cyanotic congenital heart diseases with left-to-right shunt [16].

The surgical management techniques are highly dependent on the type of TAVPR and the presence of other cardiac malformations, namely heterotaxy syndrome and hypoplastic left heart syndrome [17]. Prolonged aortic clamping time and emergency surgery are surgical factors that strongly influence patient outcome and require special postsurgical measures, as detailed in the discussion section.

## 2. SARS-CoV-2 Infection in Patients with Congenital Heart Disease

An important aspect to consider in the process of evaluating the surgical opportunity for TAPVR patients is the presence of factors that can precipitate hemodynamic and respiratory degradation [18].

Viral infections in children have long been associated with hemodynamic instability originating in systemic stress, inflammatory status, hypoxia, and metabolic imbalance [19,20,21]. MERS and SARS-CoV-2 have been linked to cardiac complications such as myocarditis, pericarditis, and arrhythmias, even in patients with no preexisting cardiac disease [22,23]. Considering the similarities between SARS-CoV-2 and the above-mentioned viruses, it is reasonable to consider, in a similar fashion, that SARS-CoV-2 infection may be associated with an unfavorable cardiac prognosis. Direct cell damage, either by cytokine storm or viral invasion, as well as ischemic injuries in the context of severe hypoxia are the underlying mechanisms of the cardiac impairment in COVID-19 infections [24]. Furthermore, the multi-inflammatory syndrome described in children exposed to SARS-CoV-2 (MIS-C) manifests itself through coagulopathies, hypotension, arrhythmias, coronary dilatation, and cardiac dysfunction [25].

Previous studies have reported arrhythmias and heart failure in patients with underlying CHD and COVID-19 infections, showing high rates of favorable evolution [26]. In a few cases, death was reported due to ventricular tachycardia or unknown causes [27,28]. To our knowledge, studies have yet to show a direct link between a specific congenital heart disease and the prognosis in the context of COVID-19 infection. Therefore, the purpose of our paper was to illustrate the prognosis of a TAPVR patient simultaneously diagnosed with COVID-19 infection, the potential risk factors taken into consideration in order to establish the best course of action, and the pre- and post-operative evolution.

## 3. Case Presentation

A 2-month-old male newborn was admitted for fever, diarrhea, and SARS-CoV-2 acute infection diagnosed through RT-PCR testing. After an initial positive response to symptomatic treatment administered in the Infectious Disease department, the patient developed respiratory distress requiring orotracheal intubation and mechanical ventilation.

At this point, the patient was referred to our cardiology clinic. He presented with altered general status, pale skin, bilateral pulmonary rales, a grade II systolic murmur, tachycardia, hypotension, distended abdomen with superficial collateral circulation, and significant hepatomegaly (the inferior liver margin was situated 4 cm below the costal rim).

The ECG revealed sinus rhythm, right QRS axis deviation with aspects of right ventricular hypertrophy, and ventricular repolarization abnormalities.

Echocardiography results revealed situs solitus, levocardia, normal systemic venous circulation, dilated inferior vena cava, atrioventricular and ventriculoarterial concordance, patent foramen ovale with right-to-left shunt; normal size and kinetics of the left ventricle, a hypertrophic, dilated right ventricle with overload indices and flattening of the interventricular septum; moderate pulmonary insufficiency, pulmonary arterial pressure (PAP) = 49 mmHg, dilated pulmonary trunk with confluent pulmonary branches, all with normal caliber, normal mitral and tricuspid valve morphology and function; common origin of the right coronary artery and circumflex artery from the right coronary sinus, venous collector that flows into the superior vena cava (SVC) and collects from three out of the four pulmonary veins; the left inferior pulmonary vein drained between the venous collector effusion and SVC effusion in the right atria, unobstructed aortic arch, no pericardial effusion.

The echocardiographic findings were highly suggestive of total anomalous pulmonary venous return, which was confirmed by thoracic computed tomography (CT).

Through thoracic CT, we found SVC with a diameter of 15 mm (Figure 1), a venous collector with a 12 mm in diameter connected to the SVC and identified at the level of a plane passing through the middle of the left superior pulmonary lobe (Figure 2). The venous structure continues superiorly with the left brachiocephalic venous trunk, which was dilated up to 12 mm (Figure 3) and inferiorly with a left pulmonary vein. The pulmonary trunk appeared shorter, with a diameter of 6 mm, right pulmonary artery = 5.4 mm and left pulmonary artery = 5.6 mm (Figure 4 and Figure 5). We have identified traits of pulmonary consolidation (Figure 6).

Biological investigations revealed microcytic hypochromic anemia, neutrophilia, lymphopenia, and hypoproteinemia. The venous blood gas test (VBG) showed SvO2 values between 96–100%, consistent with TAPVR pathophysiology.

The patient was started on inotropic support, broad spectrum antibiotic therapy, corticotherapy, and heparin for a total of 35 days, with favorable outcome.

The patient underwent surgical repair of TAPVR 4 weeks after a negative SARS-CoV-2 RT-PCR test.

The surgical procedure consisted of latero-lateral anastomosis between the venous collector and the upper part of the left atrium, closure of the atrial septal defect by direct suturing, and patent ductus arteriosus ligation. The patient required erythrocyte concentrate and plasma transfusion during surgery. CPB (cardiopulmonary bypass) exit was performed without inotropic support. The patient was extubated on the fifth postoperative day. Milrinone and diuretic infusion were administered for a total of 6 days after the surgical procedure.

A favorable result of the surgical correction was found on the control echocardiography and thoracic CT. However, the patient had remnant pulmonary lesions (left inferior lobe atelectasis), a persistent inflammatory status, and fluctuating coagulation parameters that required prolonged anticoagulation therapy and multiple courses of antibiotic therapy and anti-inflammatory treatment, alongside diuretics (furosemide 1 mg/kg/day and spironolactone 1 mg/kg.day) and captopril (0.5 mg/kg/day). Anticoagulation medication was administered for 30 days, and diuretics and angiotensin-converting enzyme inhibitors for a period of 2 months. The long-term follow-up was based on monthly cardiologic evaluation in the first year after the surgery and trimestral evaluations subsequently. We chose this considering the significant systemic burden brought on by the previous SARS-CoV-2 infection that resulted in pulmonary, hepatic, and renal distress before surgery.

## 4. Discussion

Totally aberrant pulmonary venous return is a cyanotic heart defect in which all four pulmonary veins are drained into the systemic venous circulatory system, either in the right atrium or in the tributary veins. It is identified with an incidence of 0.6–1.2 per 10,000 live births, accounting for 0.7–1.5% of congenital heart defects [29]. Prenatal diagnosis of TAPVR is rarely made, despite available knowledge and imaging techniques [30]. However, a cohort study realized by Paladini and his team, in 2017, found a significant correlation of TAPVC and other congenital heart diseases with prenatal diagnosis that was in line with postnatal findings [31].

Postnatal diagnosis is based on the clinical picture and imaging investigations. The symptoms can be severe and unresponsive to medical therapy, requiring surgical therapy. Asymptomatic forms have also been identified [8,12]. In cases without pulmonary venous obstruction, the patient has moderate cyanosis, growth delay, recurrent respiratory infections, and signs of heart failure (tachypnea, dyspnea, tachycardia, hepatomegaly) [32]. In cases with severe pulmonary obstruction, the symptoms are very pronounced—reduced exercise tolerance, cyanosis, and more pronounced signs of heart failure. Accentuated II heart sound and galloping rhythm can be detected. An infrequent heart murmur is perceived in the upper left parasternal auscultation area [29]. The electrocardiogram may find right axis deviation, signs of right ventricular hypertrophy, and tall, sharp “pulmonary“ P waves. Chest radiography can identify moderate cardiomegaly with accentuation of the pulmonary vascular pattern in the non-obstructive forms, or pulmonary edema and enlarged mediastinum without cardiomegaly in the obstructive forms. The characteristic “snowman” appearance, especially in older infants and children, is present in the supracardiac-type TAPVR [33,34]. Radiological changes can be confused with those of pneumonia, meconium aspiration, or hyaline membrane disease [35].

The echocardiographic findings included diastolic overload of the right ventricle, dilation of the pulmonary artery, and relative hypoplasia of the left heart. In the setting of pulmonary arterial hypertension, there is marked right ventricular hypertrophy, a flattened interventricular septum that exhibits paradoxical movement. Sometimes, due to inadequate transthoracic windows, other imaging investigations are required, such as contrast tomography and cardiac magnetic resonance imaging, to accurately assess the anatomy of the pulmonary veins [36].

Cardiac catheterization can provide hemodynamic details in complex cases, or palliative treatment, as is the case with percutaneous balloon atrial septostomy (Rashkind) for restrictive atrial septal defect [10,37]. However, this method is not used for diagnostic purposes.

The popular surgical approach for TAPVR repair is the excision of the common wall between the atrial septum and the coronary sinus in order to regain unobstructed blood flow to the right side of the heart [38]. The surgical repair of TAPVR is associated with various degrees of mortality and morbidity, depending on the presence of concurrent cardiac anomalies [39,40]. Heterotaxy syndrome has been proved to be a strong postoperative mortality indicator, while preoperative pulmonary venous obstruction (PPVO) greatly influences the mortality risk in patients before surgical repair [41,42,43]. In cases with PPVO, long term follow-up is warranted in order to detect postoperative obstruction lesions. Postoperative pulmonary hypertension crisis is a complication that may occur in patients with a body weight below 6 kg, heart failure, pulmonary disease, elevated PAP, PPVO, emergency surgery, and prolonged aortic cross-clamping time [44,45,46]. Prevention mechanisms include avoiding trigger factors and medical therapy with milrinone 0.5 mcg/kg/min, with sildenafil administration as part of the prevention strategy to avoid pulmonary vasoconstriction once milrinone treatment is stopped [18]. Other difficulties that may arise in these patients include postoperative arrhythmias, with early third-degree atrioventricular block and sinus node dysfunction cited as indicators of pacemaker implantation requirements in the long term. Supraventricular tachycardia has also been cited in these patients, even long after the surgical repairmen of the cardiac anomaly [47,48,49]. Low cardiac output syndrome can also prove debilitating for postsurgical recovery, as it increases the chances of acute organ damage and pulmonary infections. This is particularly worrisome considering the increasingly higher percentages of HAI infections with multiple drug-resistant bacteria [50,51]. Special considerations should also be paid to feeding protocols both before and after the surgical intervention [52]. Available guidelines elaborated by the American Society for Parenteral and Enteral Nutrition and the European Society of Paediatric and Neonatal Intensive Care proved useful in tailoring the nutritional requirements of children in postoperative care, with direct impact on the long-term prognosis [53].

Our patient’s postoperative evolution was burdened by the systemic impact of the previous SARS-CoV-2 infection. The overlap between the systemic damage caused by the COVID-19 disease, with an accentuated impact on the respiratory system, and the cellular stress associated with the surgical intervention, resulted in prolonged postoperative care and higher risk foofr hemodynamic and respiratory instability. As our patient had a persistent coagulopathy, we hypothesized that thrombotic microangiopathy (TAM), which was proved in other patients recovering from COVID-19, was the most likely cause. However, our patient did not meet the necessary criteria for TAM and received standard anticoagulation therapy indicated in patients recovering from cardiac surgery.

## 5. Conclusions

Our patient required urgent medical therapy and early surgical management, despite an initial asymptomatic status. The severe SARS-CoV-2 infection brought on both cardiac and respiratory failure, further complicating survival chances both before and after surgical repair. Our patient had no documented PPVO; however, the pulmonary strain was highly influenced by the respiratory infection, which also had a systemic impact, leading to a severe hypercoagulable state and hepato-renal decompensation. These aspects prolonged the time-before-surgery, adding insult to injury in a patient with no other anomalies other than a non-obstructive TAPVR. Emergency surgery was required in order to increase survival chances and the postoperative management of pulmonary hypertension and low cardiac output was successfully performed, proving that established protocols and the early detection of TAPVR are paramount in saving these children.

## Figures and Tables

**Figure 1 children-10-00387-f001:**
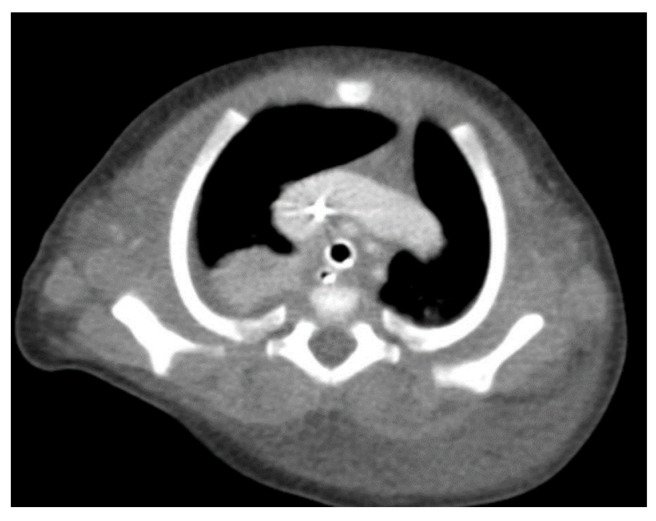
Dilated SVC.

**Figure 2 children-10-00387-f002:**
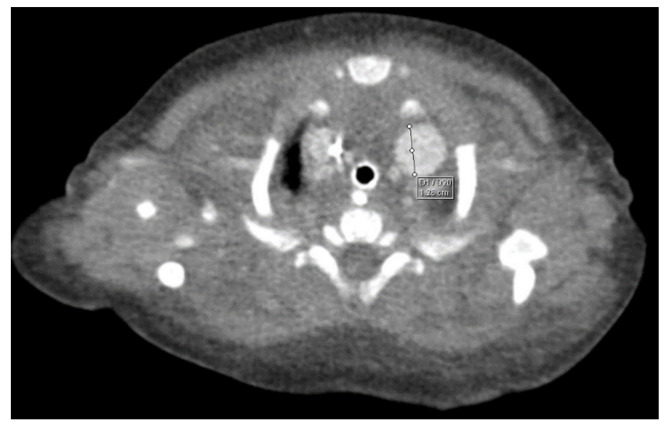
Venous collector.

**Figure 3 children-10-00387-f003:**
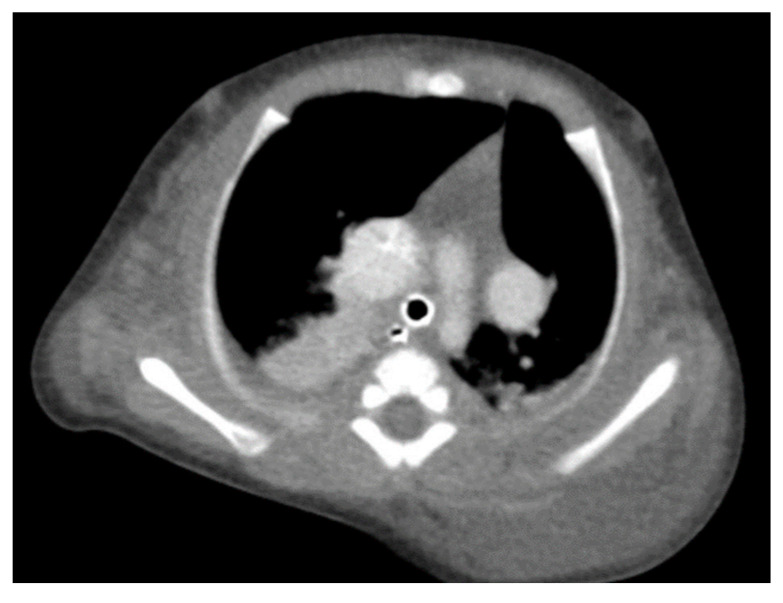
Venous brachiocephalic trunk.

**Figure 4 children-10-00387-f004:**
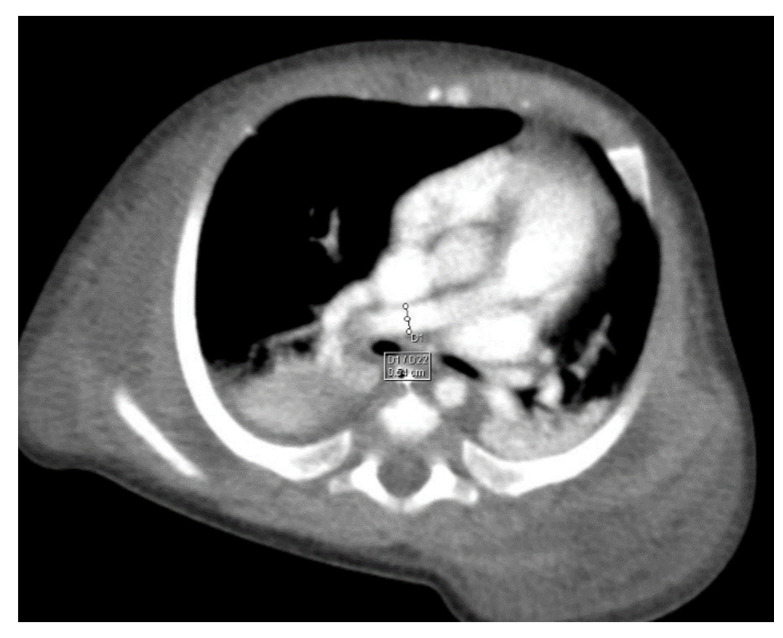
Right pulmonary artery.

**Figure 5 children-10-00387-f005:**
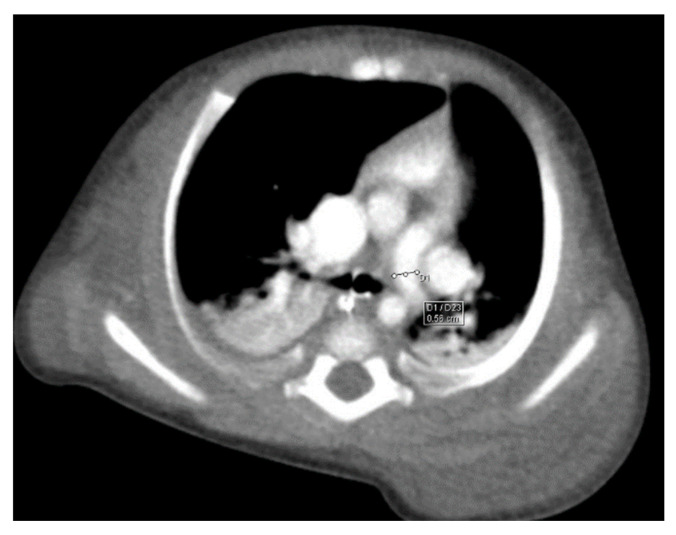
Left pulmonary artery.

**Figure 6 children-10-00387-f006:**
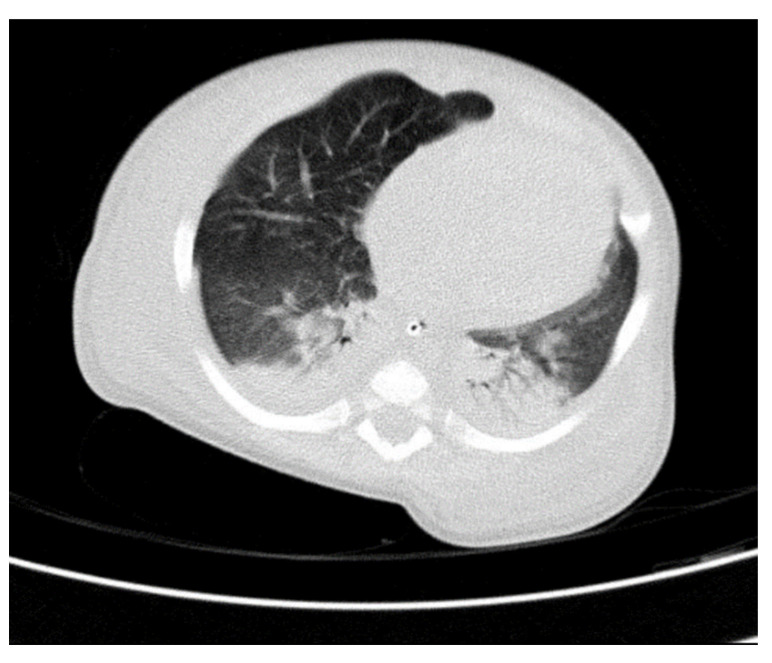
Pulmonary consolidation.

## Data Availability

All the data are available in the present paper.

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
