# Peer review of "Total Anomalous Pulmonary Venous Return in the Time of SARS-CoV-2—Case Report"

_children, 2023, doi:10.3390/children10020387_

Round 1
Reviewer 1 Report
Dear authors, this is a very interesting and difficult case. There are other published articles about total anomalous pulmonary venous return (TAPVR), however, I imagine the authors want to give importance to the SARS-Cov2. For this reason, I think that in the "Discussion" part, you should explain more extensively, how the covid contributed to an asymptomatic patient with TAPVR starting with symptoms.
Author Response
Dear Reviewer 1,
Thank your for your comments. Our interest was not to focus on SARS-Cov2 as much as it was to highlight the possible implications of being diagnosed with both and the possible high risk it entitles. Therefore, we responded as follows:
- I think that in the "Discussion" part, you should explain more extensively, how the covid contributed to an asymptomatic patient with TAPVR starting with symptoms.
We agree that we should have elaborated more on the contributions of covid to an asymptomatic patient with TAPVR. Therefore, we have added an extra paragraph in the discussion section addressing this (Lines 254-262) as well as in the introduction section (Lines 80-100).
Reviewer 2 Report
The case-report „Total anomalous pulmonary venous return in the time of 2 SARS-Cov2 – case report” highlights the role of a quick decision in front of a newborn with total anomalous pulmonary venous return (TAPVR) who underwent surgical repair of the defect with favorable outcome, despite a prior diagnosis of SARS-Cov2 infection. The authors also highlighted possible complications in the discussion regarding prognosis. I have read the paper with interest and feel that it is relevant for the area of heart surgery in children.
I suggest a few major revisions. Comments are made below regarding the article.
- The authors should clarify in the introduction the aim and objectives of the case report.
- In Introduction should be added the influence of COVID-19 disease in patients with TAPVR, as studies showed there could be consequences regarding coagulation status or myocarditis.
- In case presentation: more detailed information about the case evolution.
- In Discussion the post COVID-19 disease complication is not mentioned, please add a paragraph regarding the SARS-Cov2 infection and postop possible evolution.
- The conclusion is too long.
- References should be modified according to the journal requests for publication
- English language spelling and punctuation extend correction
Author Response
Dear Reviewer 2,
Thank your for your insightful comments. We have responded as follows:
- The authors should clarify in the introduction the aim and objectives of the case report.
We definetely agree as it was something we overlooked, therefore we have added a clear paragraph about the purpose and objectives of our study in the introduction (Lines 96-103).
- In Introduction should be added the influence of COVID-19 disease in patients with TAPVR, as studies showed there could be consequences regarding coagulation status or myocarditis.
We have added a new section in the introduction addressing the influence of covid-19 disease comparing with similar previous epidemics (MERS and SARS) as well as possible consequences (Lines 80-95).
- In case presentation: more detailed information about the case evolution.
We have added details about the respiratory, inflammatory and congulation status that required prolonged treatment (Lines 193-196).
- In Discussion the post COVID-19 disease complication is not mentioned, please add a paragraph regarding the SARS-Cov2 infection and postop possible evolution.
We have added a new paragraph in the discussion section addressing the infection and the postop evolution (Lines 257-265).
- The conclusion is too long.
With all due respect, we believe the conclusion is of a reasonable length as it summarizes the main aspects of our study and highlights the challenges. Therefore, we have left it as it was.
- References should be modified according to the journal requests for publication
We have rechecked our style compared to the journal requests for editing. As per the journal guidelines, the references can be any style as long as it is consistent. However, we have used the multidisciplinary digital publishing institute style which makes the citations in square brackets and the references with author a, b, c, title, journal name abbreviations and the year in bold.
- English language spelling and punctuation extend correction
Thank you for your observation. We have re-read and corrected all the issues we have identified.
Round 2
Reviewer 1 Report
Dear athors, congratulations on this improved version of your manuscrit. I would just add the fact that the natural history of biventricular hearts without surgical repair is very poor (50% mortality up to the third month of life and 20% survival at one year without therapeutic intervention). I would also add that the confirmed diagnosis is itself indication for surgery.
Author Response
Dear Reviewer 1,
Thank you for your kind words. We have added the observation you mentioned as we do agree that the diagnosis itself asks for the surgical therapeutic pathway.
Best regards.
Reviewer 2 Report
The case-report „Total anomalous pulmonary venous return in the time of 2 SARS-Cov2 – case report” has been improved but still some data should be added.
I suggest a few minor revisions. Comments are made below regarding the article.
- In case presentation: more detailed information about the treatment used in postoperative care regarding drugs administrated.
- English language spelling and punctuation extend correction
Author Response
Dear Reviewer 2,
Thank you for your observations. We are pleased to say that we have went through the whole paper again to check for spelling and punctuation issues and we believe we have fixed a considerable amount. Also, we have added more details regarding the postoperative care - drugs administration (Lines 201-205).
Hope we have fulfilled the requests accordingly.
Best regards.